# A Hierarchical Encoding-Decoding Scheme for Abstractive Multi-document Summarization

**Chenhui Shen**[*1,2]  **Liying Cheng**[1,3]  **Xuan-Phi Nguyen**[1,3]  **Yang You**[2]  **Lidong Bing**[†1,3]

[1]DAMO Academy, Alibaba Group, Singapore   [2]National University of Singapore
[3]Hupan Lab, 310023, Hangzhou, China
{chenhui.shen, liying.cheng, x.nguyen, l.bing}@alibaba-inc.com
youy@comp.nus.edu.sg

## Abstract

Pre-trained language models (PLMs) have achieved outstanding achievements in abstractive single-document summarization (SDS). However, such benefits may not fully extend to multi-document summarization (MDS), where the handling of cross-document information is more complex. Previous works either design new MDS architectures or apply PLMs bluntly with concatenated source documents as a reformulated SDS task. While the former does not utilize previous pre-training efforts and may not generalize well across different domains, the latter may not sufficiently attend to the intricate cross-document relationships unique to MDS tasks. Instead, we enforce hierarchy on both the encoder and decoder to better utilize a PLM to facilitate multi-document interactions for the MDS task. Across 10 MDS benchmarks from various domains, our method outperforms or is competitive with the previous best models, including those with additional MDS pre-training or with more parameters. It outperforms its corresponding PLM backbone by up to 3 ROUGE-L and is favored by humans.[1]

## 1 Introduction

Multi-document summarization (MDS) was first proposed by Barzilay et al. (1999), framed as a task of producing a single summary using a set of related documents, and has been extensively studied (Xiao et al., 2022; Song et al., 2022; Tu et al., 2022; Liu et al., 2021; Li et al., 2020; Liu and Lapata, 2019a; Fabbri et al., 2019). MDS is inherently much more complex than single-document summarization (SDS). Specifically, unlike SDS which requires the extraction of crucial information in a single article, MDS requires handling not only the contradictions from multiple sources but also the

repetitions and larger amounts of trivial information across documents (Zhou et al., 2021; Cui and Hu, 2021; Liu and Lapata, 2019a; Lebanoff et al., 2018; Li et al., 2017a, 2018; Yasunaga et al., 2017; Ganesan et al., 2010).

While MDS could be superficially converted to SDS by concatenating multiple documents into a pseudo-single document (Fabbri et al., 2019; Liu et al., 2018; Lebanoff et al., 2018), the irregular and complex multi-document information remains. As a result, it continues to pose challenges due to its large difference from the mostly coherent PLM pre-training data of continuous text segments (Baumel et al., 2018). On the other hand, many works design specialized MDS architectures (Xiao et al., 2022; Liu et al., 2021; Liu and Lapata, 2019a; Fabbri et al., 2019). However, they require large supervised MDS datasets on the scale of 100K training data (Liu et al., 2021; Liu and Lapata, 2019a; Liu et al., 2018; Fabbri et al., 2019) and may not generalize to unseen domains. Neither do they utilize the capabilities of large PLMs (Lewis et al., 2020; Raffel et al., 2020; Zhang et al., 2020) that were pre-trained on coherent data.

Given that MDS data are scarce and costly to obtain (Lebanoff et al., 2018), we propose a simple hierarchical encoding-decoding scheme to effectively fine-tune a PLM on smaller-scale MDS datasets ranging from about 2K to 45K training samples. We aim to better adapt the PLM's knowledge gained during pre-training (which is more useful to SDS due to a smaller data shift) for MDS fine-tuning in both the encoder and the decoder. In the encoder, we preserve the original attention mechanism within the same document while re-purposing "" as a document-level representation for high-level interactions. This approach ensures that intra-document tokens can be well-represented using the pre-training knowledge, while the document will still interact with one other on a higher level. In the cross-attention layers of the decoder, we impose

---

[*] Chenhui is under the Joint PhD Program between Alibaba and National University of Singapore.

[†] Corresponding author.

[1]Our code and data are fully released at https://github.com/DAMO-NLP-SG/HierEncDec.

document-level importance scaling on the attention weights, which allows each document to influence the output tokens differently but more appropriately. As a result, our method can still make use of the general language modeling capabilities of PLMs, while learning to handle the complex cross-document information during fine-tuning.

We conduct experiments on various MDS datasets (Shen et al., 2022b; Lu et al., 2020; Ghalandari et al., 2020; Fabbri et al., 2019; Wang and Ling, 2016) across a vast range of domains, namely, news, scientific literature, movie critics, peer reviews, and specific Wikipedia topics. Our experimental results show that our hierarchical scheme outperforms strong state-of-the-art models (including those with additional MDS pre-training or larger model sizes). It also consistently improves the PLM backbone on all 10 datasets by up to 3 ROUGE-L and, on the manually inspected datasets, is preferred by humans. In addition, our detailed attention, content analyses, and ablation reveal that our method ensures more coherent encoder representations for intra-document tokens, as well as obtains the wider cross-attention coverage of the different documents during decoding. In this manner, our method sufficiently utilizes the pre-training knowledge for encoding, while encouraging the decoding process to consider all documents by more proper weights, rather than over-focusing a few documents and ignoring the rest.

## 2  Related Work

**MDS Models**  Previous works have designed specific neural abstractive summarization architectures for the MDS task (Liu et al., 2021; Liu and Lapata, 2019a; Liu et al., 2018; Fabbri et al., 2019). However, it is hard to adapt the existing PLMs to these specific designs. Thus, these models have to be trained from scratch using relatively large MDS datasets on the scale of hundreds of thousands of examples. The same applies to graph-based networks (Chen et al., 2021; Li et al., 2020; Parveen and Strube, 2014; Christensen et al., 2013) that explicitly facilitate document interactions with the graphical designs. As a result, all these models may work well in the specific domain it was trained in, but may not generalize well to the smaller MDS datasets of other domains. Recent works experiment with additional MDS pre-training (Xiao et al., 2022; Puduppully and Steedman, 2022) with an existing PLM. Nevertheless, the MDS pre-training

corpus (e.g., NewSHead (Gu et al., 2020)) is much smaller in scale than the pre-training corpora, and re-training PLMs with much smaller-sized MDS corpus may lead to catastrophic forgetting (French, 1999) of the pre-trained knowledge.

**PLMs for MDS**  PLMs (Lewis et al., 2020; Raffel et al., 2020; Zhang et al., 2020) have achieved state-of-the-art performances for SDS. Some works have directly used these PLMs for MDS by concatenating multiple documents (Guo et al., 2022; Xiao et al., 2022; Shen et al., 2022a,b), including the works that focus on longer context windows (Zaheer et al., 2020; Beltagy et al., 2020). However, concatenating the content of multiple documents together prevents the PLMs from distinguishing or processing the contents differently. By essentially processing the MDS inputs as if they were a single document, the PLMs may not handle the cross-document information well. An alternative is to conduct summarization through multiple stages (Song et al., 2022; Tu et al., 2022; Ernst et al., 2022; Hokamp et al., 2020a; Li et al., 2020; Lebanoff et al., 2018; Li et al., 2017b,c; Bing et al., 2015), where salient contents are first extracted or summarized from multiple documents, then passed to a PLM for the final abstraction stage. However, the inputs to the PLM now consist of groups of incoherent sentences or phrases stitched together from multiple sources, presenting a significant distributional shift from the PLMs' original pre-training corpora.

Our method is different from the previous notable works (Xiao et al., 2022; Beltagy et al., 2020) in both that we make novel use of the start-of-document tokens as a document-level representation, as well as further utilizing these tokens during our hierarchical decoding scheme to enable cross-document interactions.

## 3  Preliminaries and Notations

We focus on adapting encoder-decoder PLMs for the MDS task. The encoder-decoder Transformer (Vaswani et al., 2017) is a common underlying architecture for PLMs frequently used for summarization tasks (Beltagy et al., 2020; Lewis et al., 2020; Zhang et al., 2020). The encoder uses self-attention to encode the source tokens, whereas the decoder uses cross-attention with source tokens and self-attention with the generated tokens for decoding. Our method targets the attention involving the

source tokens, i.e., the self-attention in the encoder and the cross-attention in the decoder.

Formally, the encoder updates the representation of each source token across all layers. The representation of the $i^{th}$ token in the $j^{th}$ layer, $\boldsymbol{h}_i^{(j)}$, can be computed as a weighted sum of the hidden states of the context tokens in the previous layer:

$$\boldsymbol{h}_i^{(j)} = \sum_{k=1}^{K} w_k \boldsymbol{V} \boldsymbol{h}_k^{(j-1)} \qquad (1)$$

where $\boldsymbol{V}$ is the value projection matrix, $K$ is the total number of input tokens, and the weight $w_k$ is obtained by a softmax calculation of the token attention scores:

$$\boldsymbol{w} = (w_0, ..., w_K) = softmax(a_0, ..., a_K) \quad (2)$$

The attention scores $\boldsymbol{a} = (a_0, ..., a_K)$ are computed with the query ($\boldsymbol{Q}$) and key matrices ($\boldsymbol{K}$):

$$\boldsymbol{a} = \boldsymbol{Q}\boldsymbol{h}_i^{(j-1)} \cdot [\boldsymbol{K}\boldsymbol{h}_0^{(j-1)}, ..., \boldsymbol{K}\boldsymbol{h}_K^{(j-1)}] \quad (3)$$

For the decoder, cross-attention is calculated between the last generated token and the source tokens. The weights calculations are similar to Equation (2) and Equation (3) except that the $\boldsymbol{Q}$ matrix takes in the representation of the last generated token (after decoder self-attention), and the $\boldsymbol{K}$ matrix takes in the representations of the source tokens from the last encoder layer.

Lastly, for multi-document encoding, we follow the common approach (Xiao et al., 2022; Shen et al., 2022b; Fabbri et al., 2019) to feed multiple inputs into the model by concatenating one document after another. With $N$ documents, the input sequence can be represented as $X = \{D_0, D_1, ..., D_N\}$, where $D_n = \{x_{n,0}, x_{n,1}, ...\}$ are the input tokens from the $n^{th}$ document.

## 4 Method

To better leverage the strong language modeling capabilities of PLMs, we propose an encoding-decoding scheme to better equip the PLMs for the MDS setting. Our design fulfills the following requirements: (1) It leverages the language modeling capability of PLMs gained during pre-training; and (2) It facilitates the model to better process cross-document information during fine-tuning with the MDS datasets.

For the first requirement, we preserve the token interactions within the same document for both the encoder and decoder. For the second requirement, we use the PLM's "" token for the hierarchical representation of the documents in the encoder, then make further use of these tokens for hierarchical attention scaling in the decoder.

### 4.1 Hierarchical Encoding Scheme

#### 4.1.1 Leveraging PLM Knowledge

To better leverage the knowledge for source token representations gained during pre-training, we apply the following modifications to the encoder: restricted intra-document full attention, and position restart.

**Restricted Intra-Document Full Attention** Since the PLM's self-attention is only applied to tokens from the same congruent source during pre-training (Lewis et al., 2020), it may not handle cross-document token representations well. Thus, to conform with the pre-training process, we restrict the self-attention for each token to only its sibling tokens within the same document (Figure 1 top left). In this way, the tokens are only allowed to see words within their own documents, which avoids leaking information or being influenced by off-context or contradictory information from other documents. Some works (Shen et al., 2022b; Fabbri et al., 2019) have directly applied full attention to tokens from all documents (Figure 2a) or use a local attention window (e.g., LED (Beltagy et al., 2020) and PRIMERA (Xiao et al., 2022)) (Figure 2b) where each token only attends to a small window of surrounding tokens. In both approaches, the token is allowed to be influenced by tokens from other documents, thus potentially causing improper representations.

**Position Restart** We restart the positional encoding for each document (Figure 1 bottom), in order to signal to our modified encoder that subsequent words are from the next document and are not continuous paragraphs of the previous document. As a result, the encoding process of each document will conform with the single-document encoding process that PLMs were pre-trained on.

#### 4.1.2 Handling Cross-Document Information

Leveraging the PLM knowledge alone can only lead to the independent processing of each document. To equip the model with capabilities to better learn cross-document information during the fine-tuning stage, we make use of global tokens for document-level information encapsulations.

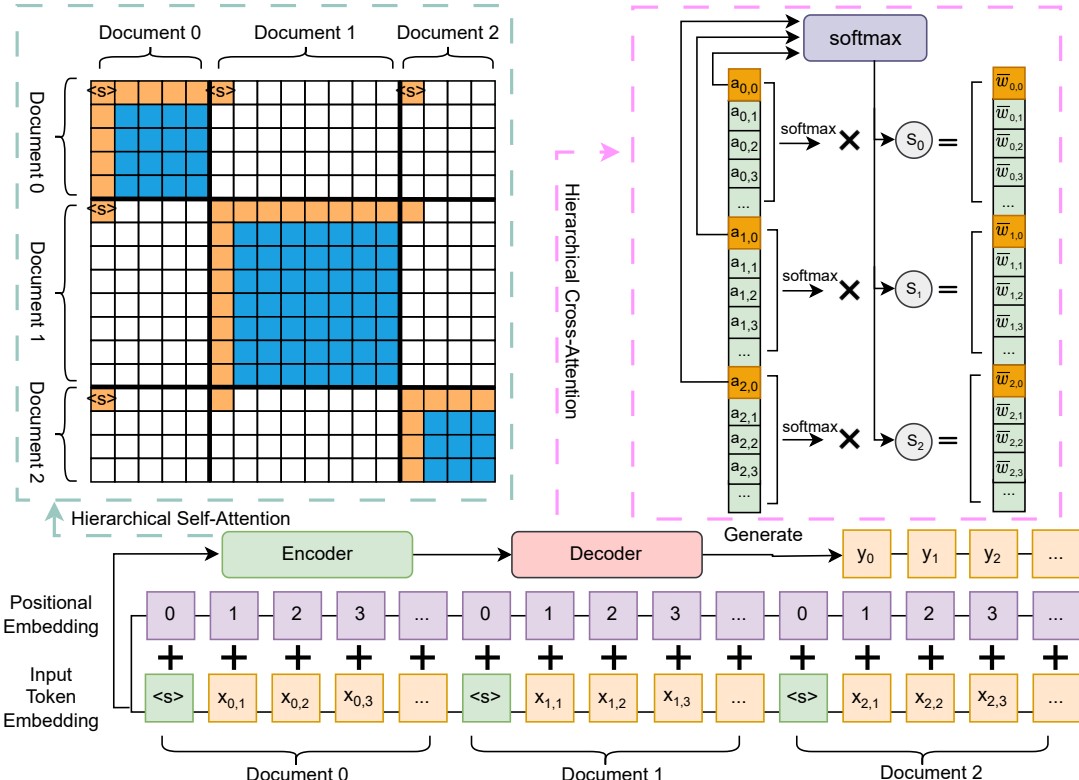

Figure 1: Our model diagram that highlights the differences compared with the PLM backbone: 1. re-start the positional embedding for each document (Section 4.1); 2. use hierarchical self-attention in the encoder (Section 4.1); 3. use hierarchical cross-attention in the decoder (Section 4.2). For "Hierarchical Self-Attention", we use bold **black** lines to indicate document borders, **blue** ■ cells for local attention, and **orange** ■ cells for document-level attention. Best viewed in color.

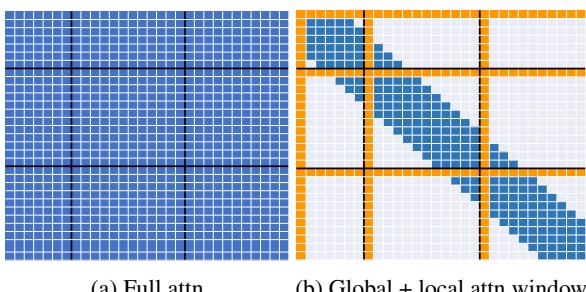

(a) Full attn     (b) Global + local attn window

Figure 2: Encoder self-attention patterns for different attention schemes. We use **black** lines to indicate document borders in the MDS setting, blue ■ cells for local attention, and orange ■ cells for global attention.

**Start-of-Document (SOD) Representations** Previous approaches have employed either a single "" token for retaining global information of the full input sequence through self-attention (e.g., LED) or have introduced multiple special "<doc-sep>" tokens between documents for the same purpose (e.g., PRIMERA). In contrast, we adopt a different strategy by utilizing the "" token to capture information from individual documents. Specifically, we position the "" token at the start of each document, serving as the

SOD token. However, instead of letting the SOD tokens attend to all tokens in the documents and act as global tokens, we *restrict their attention to the corresponding document tokens that they represent*. This is because the "" commonly resides at the sequence's outset of the pre-training inputs of PLMs, and has also been conventionally employed to encapsulate a text sequence in Transformer architectures. Consequently, the PLMs have already acquired the capability to capture the document information with the "" tokens. In this way, we leverage the "" tokens as a high-level document representation to further facilitate cross-document interactions in both the encoder and decoder (Section 4.2).

**Document-Level Attention** Unlike LED or PRIMERA which enable full attention on global tokens (shown with orange ■ blocks in Figure 2b), we only allow the SOD tokens to attend to same-document tokens and SOD tokens from other documents, as shown with the orange ■ blocks in the "hierarchical self-attention" diagram in Figure 1. This allows each SOD token to encode the informa-

tion in the same document, while only exchanging cross-document information with other SOD tokens.

## 4.2 Hierarchical Decoding Scheme

We further design a hierarchical decoder to leverage the document encapsulations in the SOD tokens, while leveraging the existing pre-training knowledge for decoding. As discussed in Section 3, the PLM's decoder first carries out self-attention with the previously generated tokens, followed by cross-attention with the source tokens. We do not modify the self-attention mechanism which is independent of the source tokens. Instead, we take advantage of the document-level information in the SOD encoder output tokens, by scaling the cross-attention attention weights towards the source tokens from the respective documents accordingly.

We illustrate the cross-attention mechanism in Figure 1 ("hierarchical cross-attention" diagram). Formally, given $N$ documents, we denote the cross-attention scores in the decoder toward *each individual document* as $\boldsymbol{a}_n = (a_{n,0}, ..., a_{n,k_n})$, where $k_n$ is the number of tokens in the $n^{\text{th}}$ document. Similar to Equation (2), we calculate the cross-attention weights for tokens within each document:

$$\boldsymbol{w}_n = (w_{n,0}, ..., w_{n,k_n}) = softmax(\boldsymbol{a}_n) \quad (4)$$

Next, we rely on the SOD tokens to decide the relative importance of each document, then scale the cross-attention weights accordingly. To do so, we first obtain the normalized scaling factor for each document as:

$$S_0, ..., S_n = softmax(a_{0,0}, ..., a_{n,0}) \quad (5)$$

where $a_{n,0} \in \boldsymbol{a}_n$ is the attention score for the SOD token in the $n^{\text{th}}$ document. We derive the normalized attention weights for tokens in each document:

$$\bar{\boldsymbol{w}}_n = (\bar{w}_{n,0}, ..., \bar{w}_{n,k}) = S_n \cdot \boldsymbol{w}_n \quad (6)$$

In this manner, our MDS decoder can better grasp the relative degree of importance of each document, while preserving relative cross-attention weights within the same document during decoding.

## 5 Experiments

## 5.1 Benchmark Datasets

We conduct experiments on a wide range of MDS datasets as follows: Multinews (Fabbri et al., 2019), WCEP (Ghalandari et al., 2020), Multi-Xscience

| Dataset | Instances | Docs | $Len_{src}$ | $Len_{tgt}$ | Train Steps |
|---|---|---|---|---|---|
| Multinews | 56K | 2.8 | 1793 | 217 | 130000 |
| WCEP | 10K | 9.1 | 3866 | 28 | 15500 |
| Multi-Xscience | 40K | 5.1 | 700 | 105 | 90000 |
| Rotten Tomatoes | 3K | 100 | 2052 | 21 | 4500 |
| MReD | 6K | 3.3 | 1478 | 120 | 10500 |
| MReD+ | 6K | 6.3 | 3069 | 120 | 10500 |
| Film | 37K | 4.5 | 777 | 92 | 85000 |
| MeanOfTransportation | 10K | 4.1 | 878 | 88 | 20000 |
| Town | 16K | 4.7 | 582 | 52 | 37000 |
| Software | 15K | 4.3 | 843 | 113 | 35000 |

Table 1: The statistics of all datasets used in this paper. The number of instances includes all training, validation, and test set examples.

(Lu et al., 2020), Rotten tomatoes (Wang and Ling, 2016), and the recently released MReD dataset (Shen et al., 2022b). These datasets are gathered from a wide range of domains, namely news, events, scientific literature, movie reviews, and peer reviews. Table 1 provides the statistics of each dataset (see download links in Appendix A).

Due to the scarcity of MDS datasets, we further compile several datasets in different domains. In the MReD peer-review dataset (Shen et al., 2022b), multiple reviews of the same research paper are used as the inputs, whereas the meta-review is used as the summary. We extend this dataset by including the corresponding rebuttals and refer to it as "MReD+". In addition, we compiled MDS datasets from four domains of Wikipedia articles[2], namely, "Film", "MeanOfTransportation", "Software", and "Town". These datasets use the Wikipedia lead section text as the summary and the individual sections as multi-document inputs.

## 5.2 Evaluated Models[3]

**BART** We fine-tune the the pre-trained "bart-large" (Lewis et al., 2020), which uses full attention (Figure 2a) for all source tokens.

**LED** LED (Beltagy et al., 2020) is a competitive baseline for long text summarization. It is directly initialized from "bart-large", but uses global-local attention to better handle long context inputs. Specifically, each local token has sliding-window attention and also attends to the global tokens, whereas global tokens attend to all source tokens. This model contains slightly more parameters due to separate $\boldsymbol{V}$, $\boldsymbol{Q}$, and $\boldsymbol{K}$ (see Equation (1) and Equation (3)) for the global tokens and local tokens.

---

[2]Original dataset from WikiAsp (Hayashi et al., 2021).
[3]The model checkpoints are detailed in Appendix B.

**LongT5** LongT5 (Guo et al., 2022) adopts summarization pre-training strategies (Zhang et al., 2020) with T5 (Raffel et al., 2020) for long context summarization. It uses transient global attention, which differs from LED in that the global tokens are obtained by summing the embeddings of tokens in k non-overlapping continuous blocks which divides the full input source. Due to resource constraints, we could only fine-tune the base model on our machines. Thus, we also provide our method results using the "bart-base" backbone of a smaller model size.

**PRIMERA** PRIMERA (Xiao et al., 2022) uses additional MDS pre-training on the NewSHead dataset on top of the LED model. It introduces a special "<doc-sep>" token in between the documents for the same function as the global token in LED. It also uses a moving context window, which may still capture tokens across two consecutive documents (see Figure 2b).

**HED (ours)** We apply our **H**ierarchical **E**ncoding **D**ecoding scheme (HED) on top of "bart-large" to obtain "BART+HED". Since HED doesn't introduce new parameters, the "BART+HED" is strictly comparable with the BART baseline. Moreover, we apply HED with the "bart-large-cnn" checkpoint, resulting in the "BART-cnn+HED" model. The "bart-large-cnn" checkpoint involves additional pre-training with an SDS dataset in the news domain (i.e., CNN/DM) over the "bart-large" checkpoint. While we do not conduct additional pre-training ourselves, we can still compare "BART-cnn+HED" with "BART+HED" to assess the knowledge transferred during the pre-training stage using an SDS dataset to the MDS downstream tasks. Moreover, this can be juxtaposed with the comparison between PRIMERA and LED, which reveals the benefits derived from extra MDS pre-training on a news dataset (i.e. NewSHead) for the MDS task.

### 5.3 Experimental Setup

We fine-tune all evaluated models discussed above with cross-entropy loss on all datasets. Following PRIMERA (Xiao et al., 2022), we use source and target truncation of 4096 and 1024 respectively. Amongst all evaluated models, LongT5 and LED can readily accept a source input of 4096 tokens. BART can only accept a maximum of 1024 tokens, so we repeatedly copy BART's positional

embeddings 4 times, similar to how LED is derived from BART by Beltagy et al. (2020). Following PRIMERA, for Multinews, WCEP, and Multi-Xscience, we truncate the end of each source document[4]. For the rest of the datasets, we follow Shen et al. (2022b) and truncate the end of the combined documents. See more explanations for our choices of truncation for different datasets in Appendix C.

We use Adam optimizer with a learning rate of $5e - 5$, and without any warm-up or weight decay. All models are trained on a single A100-80G GPU, for the same number of training steps on the same dataset as shown in Table 1. The number of training steps for each dataset is determined based on the development set losses on the "bart-large" baseline[5]. We evaluate all model outputs on the ROUGE (Lin, 2004) metric and provide the F1 values of ROUGE-1, ROUGE-2, and ROUGE-L. We also conduct human evaluations on selected benchmarks.

Additionally, we compile results directly reported by other papers (Guo et al., 2022; Song et al., 2022; Liu et al., 2022; Tu et al., 2022; Shen et al., 2022b; Pasunuru et al., 2021) in Appendix D's Table 6 to Table 10 to give a better overview of the current performances on different datasets. However, as the settings for reported models differ vastly, the results are not strictly comparable.

## 6 Results

### 6.1 Main Results

We show the ROUGE-1 and ROUGE-L results in Table 2 for all 10 datasets (see ROUGE-2 scores in Appendix Table 11). The upper section of Table 2 displays the performance of comparatively smaller models. Notably, our "BART+HED" method surpasses LongT5 across nearly all benchmarks, despite having approximately half the model size. Moving to the lower section of Table 2, our "BART+HED" model consistently exhibits improvement over the corresponding "BART" backbone, demonstrating the effectiveness of our proposed method.

In addition, "BART-cnn+HED" generally outperforms "BART+HED". This shows that our proposed method can effectively facilitate knowledge

---

[4]For instance, if there are 4 documents, each document is truncated to 4096 / 4 = 1024 tokens.

[5]We have experimented on a few datasets and observed that the same number of training steps is selected by the "bart-base" model.

| System | Size | Multinews R-1/R-L | WCEP R-1/R-L | M-XSc R-1/R-L | RT R-1/R-L | MReD R-1/R-L | MReD+ R-1/R-L | MeanOT R-1/R-L | Town R-1/R-L | Software R-1/R-L | Film R-1/R-L |
|---|---|---|---|---|---|---|---|---|---|---|---|
| LongT5 | 250M | 46.4/24.5 | 43.4/35.3 | 27.0/15.0 | 26.0/20.5 | 32.0/20.1 | **32.7**/20.6 | **41.2/33.7** | 60.2/56.7 | **37.5**/28.4 | 42.4/35.5 |
| **BART(base)+HED** | 139M | **47.1/25.0** | **44.8/36.8** | **31.9/17.7** | **26.8/20.8** | **32.2/20.6** | **32.7/20.8** | 40.6/**33.7** | **61.4/57.7** | 37.2/**28.6** | **42.8/35.9** |
| LED | 435M | 50.1/25.0 | 46.5/37.6 | 31.2/16.6 | 27.3/20.7 | 33.0/19.1 | 34.3/20.3 | 45.4/35.1 | 62.3/**58.3** | 42.1/28.8 | 44.8/35.7 |
| Primera* | 447M | 49.9/25.9 | 46.1/**37.9** | 31.9/18.0 | - | - | - | - | - | - | - |
| Primera | 447M | 49.0/25.6 | 46.2/37.4 | 31.9/18.0 | 27.4/**21.1** | 29.6/17.0 | 29.2/16.5 | 44.1/35.6 | 62.1/58.3 | 39.0/28.4 | 44.4/**36.9** |
| BART | 406M | 47.4/24.0 | 42.8/34.5 | 31.5/16.9 | 26.1/20.3 | 32.9/19.9 | 32.9/20.1 | 43.0/34.9 | 59.9/56.3 | 39.5/28.7 | 42.1/34.4 |
| **BART+HED** | 406M | 50.0/25.8 | 46.4/37.8 | 32.1/17.6 | 27.3/**21.1** | 33.9/**20.9** | 34.0/**20.7** | 43.5/35.2 | 61.9/57.7 | 40.5/**29.7** | 43.8/36.3 |
| **BART-cnn+HED** | 406M | **51.1/25.9** | **47.0**/37.6 | **34.7/18.6** | 27.6/20.5 | **34.1**/20.5 | **34.5**/20.6 | **46.1/35.4** | **62.8**/58.3 | **42.9**/29.7 | **45.9**/36.6 |

Table 2: ROUGE-1 and ROUGE-L results on 10 datasets including: Multinews, WCEP, Multi-XScience (M-XSc), Rotten Tomatoes (RT), MReD, MReD+, and 4 Wikipedia domains. *: results reported by Xiao et al. (2022).

| model | Multinews Flu | Rel | Abs | Sal | Cov | MReD Flu | Rel | Abs | Sal | Cov |
|---|---|---|---|---|---|---|---|---|---|---|
| BART | **0.510** | 0.430 | 0.475 | 0.500 | 0.480 | 0.440 | 0.480 | 0.370 | 0.355 | 0.350 |
| BART+HED | 0.490 | **0.570*** | **0.525*** | 0.500 | **0.520** | **0.550*** | 0.520 | **0.630*** | **0.645*** | **0.650*** |

Table 3: Head-to-head human evaluation scores for fluency (Flu), relevance (Rel), abstractiveness (Abs), salience (Sal), and coverage (Cov). *: statistically significant with a higher human preference for $p < 0.05$.

transfer, even for the knowledge gained with pre-training on an SDS news dataset. On the other hand, PRIMERA uses additional MDS pre-training with a news dataset on top of LED, but experiences some obvious performance drops, especially on the datasets of MReD, MRed+, and Software. As discussed previously, the MDS pre-training corpora characteristics deviate significantly from those used during the PLM's original pre-training. Thus, it is likely that additional MDS pre-training for PRIMERA on only one news dataset (i.e., New-Shead) has led to catastrophic forgetting of the other domains. As a result, the model loses some knowledge gained during its previous pre-training stage, and performs especially badly on domains that deviate significantly from the news domain.

Nonetheless, our method does not guarantee improvements with additional SDS pre-training, as evidenced by the drop in ROUGE-L for the Rotten Tomatoes. This dataset is very small, comprising of around 2K training examples. It is also unique in the sense that each sample consists of an average of 100 documents (see Table 1 "#Docs"). In this case, additional SDS pre-training may negatively impact the model's ability to handle inputs with a large number of documents, especially with very limited MDS fine-tuning data.

Another interesting observation is that LED performs relatively strongly, particularly on the WCEP and Multinews datasets. According to Wolhan-dler et al. (2022), the gold summaries of WCEP and Multinews indicate a relatively limited degree of multi-text merging, suggesting that information from a single document suffices for summarization. Consequently, although LED lacks an inherent architecture for handling cross-document information, it can still perform well because of its optimized capability to process long-context inputs.

## 6.2 Human Evaluation

We conduct human evaluations on the Multinews and MReD datasets of vastly different domains. Specifically, we randomly sample from each dataset 50 test samples to be evaluated independently by 2 random evaluators among a pool of 5 volunteers in the research field of artificial intelligence. Each evaluator is given both outputs from "BART" and "BART+HED" in a double-blind manner, and asked to rate 1 point for the better summary and 0 for the worse, while giving both 0.5 for a tie.

The evaluation criteria are (1) Fluency - the overall flow and grammar of the summary; (2) Relevance - the selection of important content from input documents; (3) Abstractiveness - the extent to which the summary consists of rephrased contents and avoids copying extensively from the input; (4) Salience - the amount of salient information included in the summary; and (5) Coverage - the minimal number of input documents required to supply sufficient information for the summary. The latter new criteria are newly proposed by us to better assess the MDS task (see more details in Appendix E).

The results are reported in Table 3. For both evaluated benchmarks, "BART+HED" outperforms "BART" overall. In terms of writing

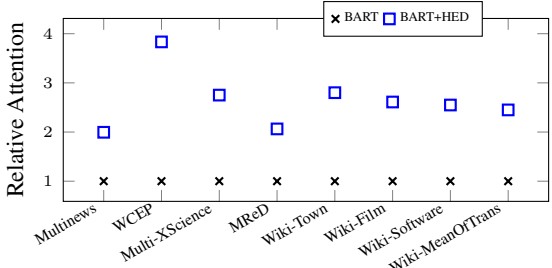
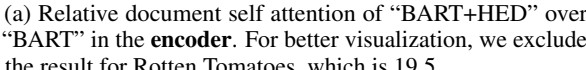
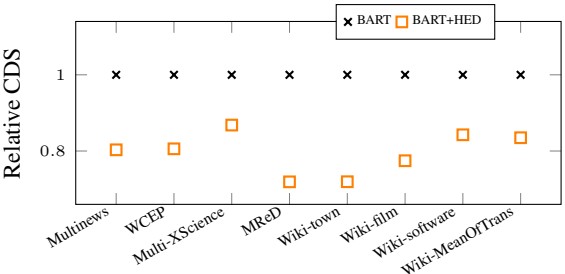

(a) Relative document self attention of "BART+HED" over "BART" in the **encoder**. For better visualization, we exclude the result for Rotten Tomatoes, which is 19.5.

(b) Relative cross-document standard deviation of "BART+HED" over "BART" in the **decoder**. We exclude the result for Rotten Tomatoes, which is statistically insignificant.

Figure 3: Document-level attention analysis. Results are statistically significant with p ≤ 0.001.

style, "BART+HED" has significantly better abstractiveness for both datasets. This suggests that "BART+HED" summarizes using its own words rather than copying long phrases and sentences verbatim from the source, thanks to the hierarchical structure that encourages explicit cross-document information handling. While higher levels of abstraction may lead to fluency degradation, "BART+HED" still performs competitively with "BART" in fluency on the Multinews dataset, and even surpasses "BART" on the MReD dataset. This indicates that despite our modifications, "BART+HED" still retains sufficient language modeling capability from "BART".

In terms of content, our method also achieves higher levels of relevance than "BART" for both datasets. This supports that our model is better at comparing input documents during decoding (also supported by Section 6.3 later) and extracting the most important information for the summary. Moreover, as MReD is from the peer-review domain, the summaries, which are meta-reviews, are highly condensed (Shen et al., 2022b) and pose a stronger challenge due to frequently conflicting input documents from disagreeing reviewers. In this dataset, "BART+HED" is superior in both salience and coverage with much higher margins, suggesting that our method is particularly effective for such complex MDS tasks.

### 6.3 Attention Analysis

To better understand the generation process, we conduct attention weights analysis on 200 test samples per dataset for both the encoder and decoder.

**Encoder Analysis** Figure 3a presents the *relative* (normalized over the "BART" baseline) attention of each source token toward its belonged document (a.k.a. self document) of "BART+HED" over the "BART" baseline in the encoder. Unsurprisingly, the self document token attention for "BART+HED" is significantly higher than the baseline across all datasets. This observation affirms that each token representation is influenced more by the coherent context of its own document while ignoring the potentially misleading or irrelevant information from other documents.

**Decoder Analysis** During decoding, we propose the **C**ross-**D**ocument **S**tandard deviation (CDS) metric (see Appendix G) to measure the standard deviation of each predicted token's normalized cross-attention weights toward different documents. We plot the *relative* CDS of our "BART+HED" model over "BART" in Figure 3b. A higher CDS indicates that the current decoding step only pays concentrated attention to a few documents, whereas a smaller CDS indicates that the attention is more evenly spread out across different documents. In Figure 3b, it is evident that our model has significantly smaller CDS values across all datasets. This shows that during the decoding process, our model pays attention more evenly across documents rather than focusing on a specific document, which helps it produce more comprehensive summaries that consider more documents.[6]

### 6.4 Content Analysis

To further validate if our method indeed produces more salient summaries, we conduct entailment-based content analyses of the generated summaries across all datasets. Inspired by Laban et al. (2022), we treat each sentence in the summary as one information unit, and then calculate the average **N**umber

---

[6] Rotten Tomatoes has near-zero absolute CDS values, which is likely due to the much higher numbers of documents per sample. These results are not included in Figure 3b due to statistical insignificance.

| System | Multinews | WCEP | M-XSc | RT | MReD | MReD+ | MeanOT | Town | Software | Film |
|---|---|---|---|---|---|---|---|---|---|---|
| BART | 0.71 | 4.47 | **0.65*** | 1.19 | 0.82 | 0.84 | 0.24 | **0.28** | 0.24 | 0.36 |
| BART+HED | **0.72** | **4.70*** | 0.46 | **1.32** | **0.83** | **1.07*** | **0.27*** | 0.27 | **0.27*** | **0.40*** |

Table 4: Sentence-normalized NED for the generated summaries. *: significantly higher with $p \leq 0.05$.

| row |  | HAE | HAD | PR | $\Delta$(R-1) | $\Delta$(R-2) | $\Delta$(R-L) |
|---|---|---|---|---|---|---|---|
| 0 | × | × | × | × | - | - | - |
| 1 | ✓ | × | × | × | +0.6 | +0.7 | +0.8 |
| 2 | ✓ | ✓ | × | × | +0.9 | +0.8 | +0.8 |
| 3 | ✓ | ✓ | × | ✓ | +1.0 | +0.8 | +0.7 |
| 4 | ✓ | ✓ | ✓ | × | +0.9 | +1.0 | +0.9 |
| 5 | ✓ | ✓ | ✓ | ✓ | +1.5 | +1.3 | +1.3 |

Table 5: Ablation. Average performance gain of using various subsets of our components as compared to the baseline BART model (row 0).

of **E**ntailed source **D**ocuments (NED) per sentence (see Appendix H). The higher the NED, the more salient the summary potentially is, since it contains information that is entailed by more documents.

As shown in Table 4, on most datasets, "BART+HED" has statistically higher NED, suggesting that our method may generate more salient summaries. One special case is Multi-Xscience (M-XSc), which uses the related work section of a paper as the target summary, and the abstract of the cited papers as input documents. Upon inspection, we discover that the summaries generated by "BART+HED" are much more abstractive and succinct (consistent with the human evaluation results on other datasets in Section 6.2), resulting in low scores below the threshold value for the entailment model used; on the other hand, the generations of "BART" copies extensively and are thus easily classified as positive entailments. When a smaller threshold is used (Appendix H), our method still outperforms the "BART" backbone in general, and the NED difference on the Multi-Xscience dataset for both models reduces to an insignificant level (p > 0.05).

### 6.5 Ablation Study

To investigate the effectiveness of each of our proposed components, we present the average performance gain from the BART baseline by using only a subset of our proposed components, namely the SOD "" token, encoder hierarchical attention (HAE), decoder hierarchical attention (HAD), and position restart (PR). Note that HAE and HAD depend on "" and HAD depends on HAE (Section 4), so they cannot be used along without their dependent components. We show the averaged results across 10 datasets in Table 5 (see full results in Appendix F).

As compared to the baseline BART (row 0), simply adding the additional "" tokens (row 1) can result in large performance gains. Next, by adding HAE in row 2, we gain some small improvements. Rows 3 and 4 show the gains of adding either HAD or PR on top of row 2. Interestingly, adding HAD or PR separately has little impact, but combining them leads to a significant gain (row 5). This shows that position restart matters more for HAD than for HAE, because it helps HAD to distinguish the different documents, while our HAE encoder already restricts the attention of each source token to the same document and is thus less affected.

## 7 Conclusion

In this paper, we work on the abstractive multi-document summarization (MDS) task. We propose a hierarchical encoding-decoding scheme, which allows effective fine-tuning of the PLM on a specific MDS task dataset without any new parameters. Our proposed scheme makes novel use of global tokens for document-level interactions in both the encoder and decoder. It can leverage the generalizing capability of PLM across a wide variety of domains. Evaluation results on 10 MDS datasets show that our approach consistently outperforms the previous best models and our PLM backbone.

### Limitations

Theoretically, our encoding-decoding scheme can reduce the space complexity of MDS summarization from $O((\sum_{n=1}^{n=N} n_k)^2)$ to $O((Max(n_0, n_1, ..., n_k)^2)$. The former is the square of the total input length from all documents, whereas the latter is simply the square of the longest document. This is because a significant amount of self-attention is no longer calculated due to our restricted intra-document full attention mechanism. However, we have not implemented such optimization in our work as the actual realization is more complicated. As a result, our work also faces the common challenge of computational resources for long document summarization. We leave the investigations for better computational efficiency to future work.

Due to the inefficient computation for long doc-

ument summarization, we focus on smaller-sized MDS datasets for PLM fine-tuning. We did not conduct experiments on the much larger datasets of WikiSum (Liu et al., 2018), Arxiv and PubMed (Cohan et al., 2018), and GovReport (Huang et al., 2021) as they are much larger in scale and require much more computational resources. Nevertheless, we believe that our encoding-decoding scheme has demonstrated consistent trends of improvement across a wide range of MDS domains.

## Ethics Statement

This paper involves the preparation of several datasets. For MReD+, we have obtained approval from the ICLR committee to use data collected from the OpenReview[7] portal. The authors of MReD (Shen et al., 2022b) already released the data for non-commercialized public usage.

For the Wikipedia domain datasets, we pair sub-portions of the articles organized by WikiSum (Liu et al., 2018) and WikiAsp (Hayashi et al., 2021), which are publicly available datasets.

## Acknowledgements

Yang You is being sponsored by NUS startup grant (Presidential Young Professorship), Singapore MOE Tier-1 grant, ByteDance grant, ARCTIC grant, SMI grant and Alibaba grant.

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

## A  Datasets Downloads

We download Multinews from `https://github.com/Alex-Fabbri/Multi-News`, WCEP from `https://github.com/complementizer/wcep-mds-dataset`, Multi-Xsceince from `https://github.com/yaolu/Multi-XScience`, Rotten Tomatoes from `https://web.eecs.umich.edu/~wangluxy/publications.html`, MReD from `https://github.com/Shen-Chenhui/MReD`. For MReD, we use the uncontrolled version of MReD since controllable summarization is not our focus. We will release the processed data of MReD+ and the 4 Wikipedia domain datasets with our code.

## B  Model Checkpoints

Our "bart-base" checkpoints are downloaded from `https://huggingface.co/facebook/bart-base`, "bart-large" from `https://huggingface.co/facebook/bart-large`, LED from `https://huggingface.co/allenai/led-large-16384`, LongT5 from `https://huggingface.co/google/long-t5-tglobal-base`, and PRIMERA from `https://github.com/allenai/PRIMER`.

In addition, depending on the model size as specified in Table 2, our "BART" was either initialized from the "bart-base" (139M) or "bart-large" (406M) checkpoints. our "BART-cnn+HED" was initialized from the "bart-large-cnn" checkpoint from `https://huggingface.co/facebook/bart-large-cnn`.

## C  Truncation Settings

We follow Xiao et al. (2022) to use a per-document truncation for Multinews, WCEP, and Multi-Xscience. This setting is reasonable for Multinews and WCEP in the news domain, because the leading sentences of a news article may account for the overall event and, thus contain more salient information for summarization. Also, for Multi-Xscience, rather than delving into specific techniques and solutions, the beginning of the abstracts may give the overall description of the task and key ideas, and thus may provide a more suitable background for summarization into the related works sections.

However, the above setting may not be equally reasonable for other domains. For the peer-review domain like MReD, the beginning sentences of reviews often give an abstract of the paper instead of discussing personal opinions that matter more

| System | R-1 | R-2 | R-L |
|---|---|---|---|
| PG-BRNN (Gehrmann et al., 2018) | 43.8 | 15.4 | 20.8 |
| Hierarchical-Transformer (Liu and Lapata, 2019a) | 42.4 | 15.3 | 22.1 |
| Hi-MAP (Fabbri et al., 2019) | 44.2 | 16.1 | 21.4 |
| GraphSum (Li et al., 2020) | 45.0 | 16.7 | 22.5 |
| Highlight-Transformer (Liu et al., 2021) | 44.6 | 15.6 | - |
| GraphSum + RoBERTa (Li et al., 2020) | 45.9 | 17.6 | 23.4 |
| CTF-DPP (Perez-Beltrachini and Lapata, 2021) | 45.8 | 15.9 | 21.0 |
| LongT5-base | 46.4 | 18.6 | 24.5 |
| **BART(base)+HED** | 47.1 | 19.4 | 25.0 |
| LongT5-large (Guo et al., 2022) | 47.2 | 18.4 | 24.2 |
| LongT5-xl (Guo et al., 2022) | 48.2 | 19.4 | 24.9 |
| BART-Long (Pasunuru et al., 2021) | 48.5 | 18.6 | 23.8 |
| BART-Long-Graph (Pasunuru et al., 2021) | 49.2 | 19.0 | 24.0 |
| PageSum-Document (Liu et al., 2022) | **51.2**‡ | **21.4**‡ | **46.9**‡ |
| PEGASUS (Zhang et al., 2020) | 47.5 | 18.7 | 24.9 |
| BigBird† | 47.8 | 19.1 | 24.7 |
| REFLECT (MLE) (Song et al., 2022) | 48.2 | 18.9 | 23.8 |
| REFLECT (CASC) (Song et al., 2022) | 49.3 | 20.0 | 24.8 |
| **BART+HED** | 50.0 | 21.0 | 25.8 |
| **BART-cnn+HED** | **51.1** | **21.5** | **25.9** |
| **BART-cnn+HED ‡** | 51.1‡ | **21.5**‡ | 46.7‡ |

Table 6: Results on Multinews. Results with citations are reported by the cited papers. ‡: results using Page-Sum's evaluation script. PageSum does not truncate the source but split all into different pages. However, we are unable to run their code due to missing environment information. **Bolded** systems: systems that use our HED method.

| System | R-1 | R-2 | R-L |
|---|---|---|---|
| BigBird† | 34.0 | 12.8 | 25.6 |
| DynE (Hokamp et al., 2020b) | 35.4 | 15.1 | 25.6 |
| UPER + LED (Tu et al., 2022) | 41.4 | 18.7 | 33.8 |
| **BART+HED** | 46.4 | 24.7 | **37.8** |
| **BART-cnn+HED** | **47.0** | **24.8** | 37.6 |

Table 7: Results on WCEP. Results with citations are reported by the cited paper. **Bolded** systems: systems that use our HED method.

to the meta-review. For Rotten Tomatoes which contains mostly single-sentence documents, per-document truncation may lead to incomprehensible broken sentences. For the Wikipedia domains, the encyclopedia-based information presented may be of equal importance regardless of their passage position. Thus for all these datasets, we follow Shen et al. (2022b) to truncate the end of the combined source.

## D  Additional Results

We gather the available results reported by other papers and present them from Table 6 to Table 10. For newer datasets such as MReD (Shen et al., 2022b), there have not been results reported except for the original paper. Also, small datasets such as Rotten Tomatoes are also less studied and we could not find very recent results. Naturally, for our newly compiled datasets from four Wikipedia

| System | R-1 | R-2 | R-L |
|---|---|---|---|
| HIERSUMM* (Liu and Lapata, 2019a) | 30.0 | 5.0 | - |
| Hi-MAP* (Fabbri et al., 2019) | 31.7 | 5.9 | - |
| POINTER-GENERATOR* (Lu et al., 2020) | 34.1 | 6.8 | - |
| BERTABS* (Liu and Lapata, 2019b) | 31.6 | 5.0 | - |
| SCIBERTABS* (Beltagy et al., 2019) | 32.1 | 5.6 | - |
| BigBird† | 31.1 | 7.0 | 16.8 |
| BART-large (Song et al., 2022) | 33.3 | 8.1 | 17.3 |
| REFLECT (MLE) (Song et al., 2022) | 33.9 | 8.1 | 17.2 |
| REFLECT (CASC) (Song et al., 2022) | 34.2 | **8.2** | 17.4 |
| BART (our baseline) | 31.5 | 7.1 | 16.9 |
| **BART+HED** | 32.1 | 6.7 | 17.6 |
| **BART-cnn+HED** | **34.7** | 7.8 | **18.6** |

Table 8: Results on Multi-XScience. Results marked with * are taken from Lu et al. (2020). Other results with citations are reported by the cited papers. Note that Song et al. (2022) may have used different hyperparameters. Their reported BART-large baseline results are much higher than ours. **Bolded** systems: systems that use our HED method.

| System | R-1 | R-2 | R-L |
|---|---|---|---|
| OPINOSIS (Ganesan et al., 2010) | 15.0 | 3.1 | 12.2 |
| MEANSUM (Chu and Liu, 2018) | 15.8 | 1.9 | 12.3 |
| CONDASUM (Amplayo and Lapata, 2019) | 22.5 | 7.7 | 18.5 |
| DENOISESUM (Amplayo et al., 2020) | 21.3 | 4.6 | 16.3 |
| PLANSUM (Amplayo and Lapata, 2020) | 21.8 | 6.2 | 17.0 |
| BigBird† | 19.3 | 2.5 | 14.0 |
| **BART+HED** | 27.3 | **9.7** | **21.1** |
| **BART-cnn+HED** | **27.6** | 9.4 | 20.5 |

Table 9: Results on Rotten Tomatoes. Results with citations are reported by the cited paper. **Bolded** systems: systems that use our HED method. Unfortunately, we cannot find other more recent reported results on this dataset.

| System | R-1 | R-2 | R-L |
|---|---|---|---|
| BART-cnn * | 33.3 | 8.6 | 19.7 |
| BigBird† | 33.0 | 9.4 | 20.6 |
| **BART+HED** | 33.7 | 9.3 | **20.5** |
| **BART-cnn+HED** | **34.1** | **9.5** | 20.5 |

Table 10: Results on MReD. *: Reported by Shen et al. (2022b). **Bolded** systems: systems that use our HED method.

domains, there are no previously reported results.

Note that the additionally reported results are meant for a better understanding of the current best performances only, because many of the reported models are not directly comparable to our setting (or comparable with each other) due to different model sizes and experimental settings. For instance, LongT5-large (Guo et al., 2022) uses 780M parameters, and LongT5-xl (Guo et al., 2022) uses 3B parameters, which are not on the same scale as the BART models. Moreover, multi-stage summarization models use multiple PLMs. For instance, REFLECT (Song et al., 2022) uses 2 RoBERTa-base and BART-large, whereas UPPER + LED (Tu et al., 2022) uses GPT and LED for the extraction and abstraction stages respectively. Not all the models use the same source truncation either. In one extreme case, PageSum (Liu et al., 2022) uses the full source input with multiple encoding stages. Lastly, certain models didn't use pre-training data and are trained directly on the MDS dataset, such as Hi-MAP (Fabbri et al., 2019) and Highlight-Transformer (Liu et al., 2021).

In addition, we also experiment with BigBird (Zaheer et al., 2020) with the same experimental setup in Section 5 on 5 datasets. Unfortunately, we could only find BigBird's checkpoints for summarization trained on one of the following datasets: ArXiv (Cohan et al., 2018), PubMed (Cohan et al., 2018), or BigPatent (Sharma et al., 2019). We use the Huggingface checkpoint at `https://huggingface.co/google/bigbird-pegasus-large-arxiv`. As shown by Table 6, 7, 8, 9, and 10, BigBird can still per-

form reasonably on larger datasets such as Multi-news, Multi-Xscience, and WCEP, but it lags behind other more competitive models. However, for small datasets such as Rotten Tomatoes, there is serious performance degradation. Nevertheless, given that ArXiv's domain is quite close to MReD, BigBird can reach competitive performance on this dataset.

# E Human Evaluation

We define 2 evaluation criteria specifically for the MDS setting: salience and coverage. MDS involves distilling important information from multiple input documents. Thus, we use salience and coverage to measure how the summary makes use of the input documents from 2 perspectives.

First, we treat each sentence in the summary as an information unit and count the total number of input documents that support the information units for the whole summary. If the summary includes consensus or opinions commonly agreed on by most input documents, it should have a larger document count as compared to another summary that overly focuses on trivial details or contains extensive hallucinations. We normalize the total count by the total number of sentences in the summary and regard it as the salience of the summary.

Second, we measure the minimum number of input documents required to generate the corre-

| System | Size | Multinews R-2 | WCEP R-2 | M-XSc R-2 | RT R-2 | MReD R-2 | MReD+ R-2 | MeanOT R-2 | Town R-2 | Software R-2 | Film R-2 |
|---|---|---|---|---|---|---|---|---|---|---|---|
| LongT5 | 250M | 18.6 | 22.5 | 6.5 | 9.4 | 8.6 | 9.3 | **23.1** | 45.7 | 18.5 | 24.8 |
| **BART+HED** | 139M | **19.4** | **23.9** | **6.8** | **9.6** | **9.5** | **9.4** | 23.1 | **47.1** | **18.8** | **25.2** |
| LED | 435M | 20.5 | 24.7 | 7.3 | 9.9 | 8.5 | 9.4 | 24.7 | 47.9 | 19.2 | 25.0 |
| PRIMERA* | 447M | 21.1 | **25.3** | 7.4 | - | - | - | - | - | - | - |
| PRIMERA | 447M | 20.5 | 24.8 | 7.4 | **10.3** | 7.3 | 7.3 | 25.4 | **47.9** | 19.2 | **26.5** |
| BART | 406M | 18.3 | 21.6 | 7.1 | 9.4 | 8.8 | 9.1 | 24.4 | 45.6 | 18.8 | 23.2 |
| **BART+HED** | 406M | 21.0 | 24.7 | 6.7 | 9.7 | **9.7** | **9.7** | 24.7 | 47.7 | 20.0 | 25.7 |
| **BART-cnn+HED** | 406M | **21.5** | 24.8 | **7.8** | 9.4 | 9.5 | **9.7** | **24.8** | 47.8 | 19.9 | 25.8 |

Table 11: ROUGE-2 results on 10 datasets including: Multinews, WCEP, Multi-XScience (M-XSc), Rotten Tomatoes (RT), MReD, MReD+, and 4 Wikipedia domains. *: results reported by Xiao et al. (2022).

| System | Multinews R-1/R-2/R-L | WCEP R-1/R-2/R-L | M-XSc R-1/R-2/R-L | RT R-1/R-2/R-L | MReD R-1/R-2/R-L |
|---|---|---|---|---|---|
| BART | 47.4/18.3/24.0 | 42.8/21.6/34.5 | 31.5/7.1/16.9 | 26.1/9.4/20.3 | 32.9/8.8/19.9 |
| +  | 48.7/19.6/24.9 | 44.5/23.3/36.3 | 31.4/6.4/17.3 | 26.6/9.6/20.6 | 32.9/9.0/20.1 |
| +  + HAE | 49.3/20.2/25.2 | 45.1/24.0/36.8 | 32.3/6.9/17.2 | 26.7/9.6/20.6 | 32.7/8.8/20.2 |
| +  + HAE + PR | 49.1/20.3/25.4 | 44.9/23.4/36.5 | 32.0/6.4/17.1 | 27.1/9.8/20.7 | 33.0/8.6/19.5 |
| +  + HAD + HAD | 48.8/20.0/25.1 | 45.7/24.5/37.4 | 32.1/7.3/17.4 | 26.4/9.3/20.5 | 33.4/9.4/20.6 |
| +  + HAE + HAD + PR | 50.0/21.0/25.8 | 46.4/24.7/37.8 | 32.1/6.7/17.6 | 27.3/9.7/21.1 | 33.9/9.7/20.9 |

| System | MReD+ R-1/R-2/R-L | MeanOT R-1/R-2/R-L | Town R-1/R-2/R-L | Software R-1/R-2/R-L | Film R-1/R-2/R-L |
|---|---|---|---|---|---|
| BART | 32.9/9.1/20.1 | 43.0/24.4/34.9 | 59.9/45.6/56.3 | 39.5/18.8/28.7 | 42.1/23.2/34.4 |
| +  | 33.4/9.3/20.6 | 42.6/24.4/35.1 | 61.1/47.2/57.7 | 39.8/19.6/29.4 | 43.0/25.3/36.0 |
| +  + HAE | 33.4/8.9/20.0 | 42.8/24.2/34.7 | 61.2/46.6/57.5 | 40.4/19.6/29.4 | 43.9/25.5/36.2 |
| +  + HAE + PR | 33.3/9.0/19.7 | 43.3/24.6/35.1 | 61.5/47.4/57.7 | 40.0/19.6/29.4 | 43.7/25.6/36.2 |
| +  + HAE + HAD | 33.6/9.5/20.7 | 44.5/24.8/35.5 | 60.1/46.5/56.8 | 38.8/19.2 29.0 | 43.3/25.3/36.1 |
| +  + HAE + HAD + PR | 34.0/9.7/20.7 | 43.5/24.7/35.2 | 61.9/47.7/58.1 | 40.5/20.0/29.7 | 43.8/25.7/36.3 |

Table 12: Detailed ablation results.

sponding summary. The summary should represent all input documents as much as possible, to provide holistic perspectives. We also normalize this count over the total number of sentences in the summary and regard it as the coverage of the summary.

For instance, if document #1 contains information A and B, document #2 contains information A and C, document #3 contains information C and D, and the summary contains information A, B, and C in 3 sentences, then the salience of the summary is 1.67, and the coverage is 0.67. For salience, because A is supported by 2 documents, B is supported by 2 documents, and C is supported by 1 document, it is a total of 5 documents divided by 3 summary sentences. For coverage, using documents #1 and #3 alone is sufficient for the summary, and hence it is 2 documents, divided by 3 summary sentences.

For each pair of summaries, we first convert the score from each annotator to a head-to-head score, before calculating the final average score between the 2 annotators.

## F Ablation

In this section, we show the detailed ablation results for all 10 datasets in Table 12. It can be seen that consistent improvements can be achieved by adding the HierEnc. Next, implementing decoder attention scaling with position restart can make further use of the "" tokens to achieve the best results.

## G Cross-Document Standard Deviation

We provide details of the Cross-Document Standard Deviation (CDS) calculation below. First, during the model's inference stage, we can easily obtain the normalized cross-attention weights of each decoded token toward the source tokens in all attention heads and layers. We average these weights for all attention heads and layers for each individual token, then aggregate the weights for the tokens belonging to the same document. Thus, the cross-attention for the $n^{th}$ document can be calculated as:

$$w_{D_n} = \sum_{k=1}^{n_K} w_{n,0}, ..., w_{n,k}, \qquad (7)$$

| System | Multinews | WCEP | M-XSc | RT | MReD | MReD+ | MeanOT | Town | Software | Film |
|--------|-----------|------|-------|-----|------|-------|--------|------|----------|------|
| BART | 0.81 | 5.22 | **0.85** | 3.69 | **1.20** | 1.43 | 0.46 | 0.40 | 0.42 | 0.62 |
| BART+HED | **0.82** | **5.45\*** | 0.74 | **4.42\*** | 1.19 | **1.68\*** | **0.48** | **0.42** | **0.45\*** | **0.65\*** |

Table 13: Sentence-normalized NED for the generated summaries using entailment threshold = 0.2. \*: significantly higher with p ≤ 0.05.

where $n_K$ is the total number of tokens in the $n^{th}$ document. Next, we normalize the cross-attention for all documents as:

$$\{\hat{w_{D_0}}, ...\hat{w_{D_n}}\} = softmax\{w_{D_0}, ..., w_{D_n}\} \quad (8)$$

Then, we can obtain the CDS score for the $i^{th}$ generated token over the source documents by:

$$CDS_i = Std\{\hat{w_{D_0}}, ...\hat{w_{D_n}}\} \quad (9)$$

Note that for ease of denotation, we do not specify the $i$ value in Equation (7) and Equation (8), but the set of weights $\{w_{n,0}, ..., w_{n,k}\}$ are specific to the $i^{th}$ token. Finally, we can obtain a single CDS value for each test instance by averaging the CDS values for all generated tokens.

We can calculate for both the "BART+HED" and "BART" models over the same test input, then divide the CDS value of "BART+HED" by that of "BART" to obtain the relative CDS value. Our relative CDS value provided Figure 3b is the average of all 200 analyzed examples.

## H   Content Analysis

Inspired by Laban et al. (2022), we use a natural language entailment (NLI) model[8] to evaluate which documents each generated sentence in the summary makes use of.

Specifically, the code of Laban et al. (2022) can output a score for the entailment of each sentence towards an input document from a range of -1 to 1, where -1 indicates total contradiction, and 1 indicates perfect entailment. We use a threshold of 0.5[9], and if one generated sentence has a higher entailment score than the threshold with one document, we consider this sentence relies on the information of that particular document.

In this way, for each sentence, we can count the total number of documents entailed. We average this number across all summaries generated by each system on the same dataset, and name this metric "NED". A higher NED number indicates that the summary has on average more entailed documents for each sentence. Thus, it is likely that the summary includes more crucial information that is agreed upon by more documents.

In addition, we present the NED using threshold = 0.2 in Table 13. The general trend is similar to using threshold = 0.5 (Table 4), that our method produces summaries with higher NED over most datasets. Moreover, with a lower NED requirement, we can see that the gap between "BART" and "BART+HED" on Multi-Xscience reduces to an insignificant level. This supports our hypothesis that "BART+HED" has lower NED on this particular dataset due to its much more abstractive generations. The NLI model may mistakenly classify the much more abstractive summaries as not entailed to any document, and may not indicate that the salience of "BART+HED" is worse than "BART".

---

[8]Following Laban et al. (2022), we use the Huggingface checkpoint https://huggingface.co/tals/albert-xlarge-vitaminc-mnlias our NLI model.

[9]Although from our experience with the code, a score as low as 0.2 can correctly identify a correct entailment, we use a higher threshold just to be sure not to mistakenly include any wrong documents.