# OpenReview forum: "A Hierarchical Encoding-Decoding Scheme for Abstractive Multi-document Summarization"
_EMNLP/2023/Conference — EMNLP 2023 Findings_

### Official Review · Reviewer_62Tb · 2023-08-03

**Soundness:** 2

**Excitement:**

3: Ambivalent: It has merits (e.g., it reports state-of-the-art results, the idea is nice), but there are key weaknesses (e.g., it describes incremental work), and it can significantly benefit from another round of revision. However, I won't object to accepting it if my co-reviewers champion it.

**Paper Topic And Main Contributions:**

The paper proposes a new model which uses Hierarchical context for the task of MDS.The paper highlights the limitations of existing approaches, where some either introduce new MDS architectures or naively treat MDS as a concatenated SDS task. The former might not leverage the full potential of pre-training, while the latter may not effectively capture the intricate inter-document relationships inherent to MDS.

**Questions For The Authors:**

How it is different from a combination of PRIMER and LED?
Rouge scores computed by a same script or scores carried forward?
How does the proposed model perform over other metrics?

**Reasons To Accept:**

The paper proposes a novel method that enforces hierarchy on both the encoder and decoder, aiming to better leverage PLMs for facilitating interactions between multiple documents in the context of MDS.
The paper claims to have achieved promising results across 10 MDS benchmarks from various domains. The method reportedly outperforms or at least competes well with the best-performing models in the field, including those with additional MDS pre-training or with more model parameters.
The arguments are sound and the paper is written well.

**Reasons To Reject:**

The methodology looks like an extension of PRIMERA only. The method is not very convincing.
It is also unclear if all baselines and the proposed architecture are evaluated on same Rouge metric or reported results from other papers are carried forward.
How does your system perform over other metrics? for example BERT or BARTScore, QA based metrics? Rouge alone can not suffice as a improvement.

**Reproducibility:**

3: Could reproduce the results with some difficulty. The settings of parameters are underspecified or subjectively determined; the training/evaluation data are not widely available.

**Reviewer Confidence:**

5: Positive that my evaluation is correct. I read the paper very carefully and I am very familiar with related work.

---

> ### Author Rebuttal · Authors · 2023-08-29
>
> We thank the reviewer for the thoughtful review! Let us clarify and address your concerns as follows:
>
> * _Our method is not similar to "combining PRIMERA and LED"._
>
> ‒ We seek to clarify that PRIMER's methodology is to use carefully designed MDS __pre-training objectives__ (lines 363-365) on top of a minimally modified LED model. So we can say that PRIMERA is close to an LED with additional MDS pre-training.
>
> ‒ Our work differs from PRIMERA by directly equipping BART with document-level understandings __without additional pretraining__.
>
> ‒ Moreover, both PRIMERA and LED use __sliding local attention windows__ (line 241-248, and Fig. 2b) on the encoder __without any decoder modification__, where ours induces __document-level representation__ in the encoder (Fig.1 top left), and is the __first__ to scale the same decoder's cross attention according to documents (Fig.1 top right).
>
> We will improve our writing to highlight all of the above in the paper.
>
> * _We believe our method is as convincing as it could be for this kind of research work,_ that is:
>
> ‒ We evaluate our work using __both Rouge and human evaluations__ for our results, which is consistent with the __standard process in summarization literature__ ([Guo et al., 2022], [Xiao et al., 2022], [Beltagy et al., 2020]). The results demonstrate that our method consistently outperforms previous baselines and establishes the SOTA.
>
> ‒ In addition, for __all 10 datasets__, we provide detailed __attention analysis__ on the encoder and decoder to show how our method differs (Sec 6.3),  use __content analysis__ to show our method generally produces more salient summaries (Sec 6.4), and also provide a detailed __ablation analysis__ to examine each of our proposed components.
>
> * _Whether all baselines are evaluated on the same ROUGE:_
>
> Yes, we run experiments and use the same evaluation script on all baselines. We explicitly highlight places where we __additionally__ include reported values from cited papers (e.g., Tab 1's caption with * sign, Appendix D lines 976-978).
>
> * _Evaluation results with BARTScore, BERTScore, and QA metric:_
>
> We seek the reviewer's understanding that many summarization papers evaluated on Rouge alone or with some human evaluation are accepted by the NLP community ([Guo et al., 2022], [Xiao et al., 2022], [Beltagy et al., 2020]). We provide additional results for the requested metrics below, which all show that our method is the SOTA in general:
>
>
> BERTScore (rescaled)
> |          model                |   Multinews  |   WCEP    |   M-XSc      |  RT           |   MReD    |
> | ---------------------   | -----------   | -----------  | -----------   | ----------- | ----------- |
> | LED                             |     0.248     |    0.484      |      0.060    |     0.317     |     0.110   |
> | PRIMERA                    |     0.243     |     0.477     |      0.025    |     0.330     |    0.061    |
> | BART                           | 0.222        | 0.451         | 0.077         | 0.323         | 0.138        |
> | BART+HED _(ours)_        | 0.258        | __0.495__  | __0.103__  | __0.330__ | __0.168__ |
> | BART-cnn+HED _(ours)_ | __0.262__ | 0.490         | 0.055         | 0.321         | 0.134        |
>
> BARTScore (F1)
>
> |          model                |   Multinews  |   WCEP     |   M-XSc      |  RT           |   MReD    |
> | ---------------------      | -----------   | -----------   | -----------    | ----------- | ----------- |
> | LED                              | -4.47          | -4.11        | -5.84        |   -5.01        | -5.66         |
> | PRIMERA                     | -4.45          | -4.13         | -5.93         | __-4.93__ | -5.86        |
> | BART                            | -4.64          | -4.35         | -5.84         | -4.59        | -5.66         |
> | BART+HED _(ours)_        | -4.39        |         -4.11  | __-5.46__  | -4.99         | __-5.31__ |
> | BART-cnn+HED _(ours)_ | __-4.38__   | __-4.08__ | -5.97         | -5.06         | -5.53         |
>
> QuestEval:
>
> Unfortunately, this metric takes an excessively long time to run, as it uses 2 generation models. Multinews and Multi-Xscience are the 2 largest datasets in our study. Each dataset requires 10+ GPU hours for each test output file of 5.5k+ examples, so 5 models with 3 runs require 150+ GPU hours per dataset, which is infeasible to complete in a short time.
>
> |          model                |   WCEP      |  RT           |   MReD    |
> | ---------------------      | -----------   | ----------- | ----------- |
> | LED                              | 0.481          |   0.370       | 0.319       |
> | PRIMERA                     | 0.478          |     0.364     | 0.309        |
> | BART                           | 0.462          | 0.360        | 0.314       |
> | BART+HED _(ours)_      | __0.484__   | __0.371__   | __0.324__ |
> | BART-cnn+HED _(ours)_  | 0.483        | __0.371__    |  0.318      |
>
> References:
>
> [Guo et al., 2022] LongT5: Efficient Text-To-Text Transformer for Long Sequences (NAACL 2022)
>
> [Xiao et al., 2022] PRIMER: Pyramid-based Masked Sentence Pre-training for Multi-document Summarization (ACL 2022)
>
> [Beltagy et al., 2020] Longformer: The Long-Document Transformer (arxiv 2020)

---

### Official Review · Reviewer_VCWy · 2023-08-06

**Typos Grammar Style And Presentation Improvements:** 052 much large
the tone of the prose …
**Soundness:** 4

**Excitement:**

3: Ambivalent: It has merits (e.g., it reports state-of-the-art results, the idea is nice), but there are key weaknesses (e.g., it describes incremental work), and it can significantly benefit from another round of revision. However, I won't object to accepting it if my co-reviewers champion it.

**Paper Topic And Main Contributions:**

The authors introduce a method for multi document summarization which enforces a hierarchy upon both encoder and decoder components of pre-trained language models.
The authors work with transformer-based PLMs, and the approach uses the special token <s> at the beginning of each document to provide a document-level representation in the encoder, which is used to produce a document-level weighting that is applied to partitions of the attention weights in the decoder. In addition, the encoder attention is masked so that tokens within a document can only attend to other tokens from the same document. Decoder attention is scaled by document-level weights that the authors claim reflect the relative contribution of each document to the summary.


**Reasons To Accept:**

- Very thorough evaluation including a broad set of evaluation datasets as well as human evaluation
- clear presentation of results and inclusion of ablation study of components
- straightforward methodology that should be easy to replicate


**Reasons To Reject:**

- minimal scientific contribution in terms of methodology
- incremental gains over SOTA results

**Reproducibility:**

5: Could easily reproduce the results.

**Reviewer Confidence:**

4: Quite sure. I tried to check the important points carefully. It's unlikely, though conceivable, that I missed something that should affect my ratings.

---

> ### Author Rebuttal · Authors · 2023-08-29
>
> We thank the reviewer for the thoughtful review and suggestions! Let us clarify and address your concerns as follows:
>
> * _"minimal scientific contribution in terms of methodology"_
>
> Our methodology is a significant contribution to the realm of MDS in the following ways
>
> ‒ Unlike previous work, which either trains hierarchical models from scratch or applies PLM without considering the MDS structures, ​our method readily adapts an __existing pre-trained model__ to achieve __SOTA results__ without requiring a complete re-training job or additional MDS pre-train data.
>
> ‒ Therefore, our method saves computational costs of approx. __25k V100 GPU hours__ [Liu et al., 2019] (__~100k USD__ on AWS compute service) with significantly less compute and training data.
>
> ‒ Moreover, in terms of technical details, we are the __first__ to scale the same decoder's cross attention according to documents, and we also differ from previous works by using encoder intra-document attention (as contrasted to the sliding window) to distill document-level representations. Our detailed analysis shows that our method leads to more __coherent__ encoder representations of the same-document tokens and __wider cross-attention coverage__ of different documents  (Sec 6.3 and 6.4).
>
> * _"Incremental gains over SOTA results"_
>
> It can be observed in Tab 1 that a SOTA model may be stronger in some datasets but weaker in others, yet our method generally outperforms all models and shows non-trivial improvements (&GreaterEqual; 1 Rouge-1 point) over __any single__  previous SOTA model. Using Rouge-1 as an example:
>
> ‒ Our method outperforms PRIMERA on Multinews by 2.1, on M-Xsc by 2.8, and on MReD by 4.5, MReD+ by 5.3, MeanOT by 2.0, Software by 3.9, and Film by 1.5;
>
> ‒ Our method outperforms LED on Multinews by 1.0, M-XSc by 3.5, MReD by 1.1, and Film by 1.1;
>
> ‒ Our method (with only __half the model size__) outperforms LongT5 on WCEP by 1.4, M-Xsc by 4.9, and Town by 1.2;
>
> * _Presentation_
>
> We appreciate the reviewer's suggestions!
>
> ‒ We will revise 052 as __"...MDS architectures require MDS supervised dataset on the scale of 100k examples to train from scratch for reasonable performance"__  (citing the works in lines 109-110)
>
> ‒ We will revise 063 as __" ... more suitable for SDS because ...."__  and elaborate with reasons in lines 231-233, and lines 246-248​. We also include [Li et al., 2020] and[Liu and Lapata, 2019] for citations.
> ​​
>
> References:
>
> [Liu et al., 2019] RoBERTa: A Robustly Optimized BERT Pretraining Approach (arxiv 2019)
>
> [Li et al., 2020] Leveraging Graph to Improve Abstractive Multi-Document Summarization (ACL 2020)
>
> [Liu and Lapata, 2019] Hierarchical Transformers for Multi-Document Summarization (ACL 2019)

---

### Official Review · Reviewer_MRbH · 2023-08-12

**Soundness:** 3

**Excitement:**

4: Strong: This paper deepens the understanding of some phenomenon or lowers the barriers to an existing research direction.

**Paper Topic And Main Contributions:**

Typically Multi-Document Summarization (MDS) has been handled in two ways: A] Proposing a completely new MDS architecture or B] Use PLMs and treat MDS as SDS task by concatenating the documents. The first approach cannot leverage existing pretraining information and the second cannot leverage intricate details unique to MDS task. The paper proposes a modification to the existing encoder-decoder models that better facilitates multi-document summarization while being able to leverage existing pre-training information.

**Questions For The Authors:**

[A] PRIMERA's reported numbers seem to be lower than LED on some of the datasets. However, PRIMERA uses LED as base so why is there such a discrepancy?

**Reasons To Accept:**

1. The proposed approach is simple.
2. The performance numbers are good.

Based on the provided clarification:
1. > Regardless, our method is not similar to Guo et al. in that they use sliding window attentions closer to Longformer and BigBird, and used 100k examples to pre-train the model for code tasks; on the other hand, ours seeks to equip the pre-trained BART with document-level attentions for the MDS domain with as little as 3K examples during fine-tuning.
2. > In the context of MDS, our method is not only novel but also achieves the SOTA and is superior to other methods that attempt to modify the attention mechanism, such as LongT5 and LED.

**Reasons To Reject:**

Limited Novelty, similar approaches have been applied in other context. For example, [Guo et. al., 2023] introduce the concept of bridge and memory tokens in code, where the attention across various code blocks is restricted via these tokens. However, one could consider this application to MDS as novel.

# References
[Guo et. al., 2023], LongCoder: A Long-Range Pre-trained Language Model for Code Completion, ICML 2023

**Reproducibility:**

4: Could mostly reproduce the results, but there may be some variation because of sample variance or minor variations in their interpretation of the protocol or method.

**Reviewer Confidence:**

4: Quite sure. I tried to check the important points carefully. It's unlikely, though conceivable, that I missed something that should affect my ratings.

---

> ### Author Rebuttal · Authors · 2023-08-29
>
> We thank the reviewer for the thoughtful review! Let us clarify and address your concerns as follows:
>
> * [Guo et al., 2023] is public on June 26, 2023, while the EMNLP deadline was June 23, 2023. Our paper is also public before the EMNLP anonymity period (May 23, 2023). We seek the reviewer's understanding that a comparison with [Guo et al. 2023] is therefore **not warranted**. Regardless, our method is not similar to Guo et al. in that they use sliding window attentions closer to Longformer and BigBird, and used 100k examples to pre-train the model for code tasks; on the other hand, ours seeks to equip the pre-trained BART with document-level attentions for the MDS domain with as little as 3K examples during fine-tuning.
>
> * In the context of MDS, our method is not only novel but also achieves the SOTA and is superior to other methods that attempt to modify the attention mechanism, such as LongT5 and LED.
>
> * PRIMERA has lower results than LED on the smaller out-of-domain datasets. This is most likely because PRIMERA's additional MDS pre-training in the news domain leads to catastrophic forgetting of the general knowledge gained during the original pre-training stage (more details in lines 437-451). This may not be much of an issue if there is still sufficient training data (i.e. Multi-Xscience with 40K examples), but for MReD with only 6K examples, it struggles to keep on par.  ​This behavior can also be observed from our BigBird results (using the checkpoint pre-trained in the scientific domain) shown in the appendix (lines 1007-1024). While BigBird has acceptable performance on large out-of-domain datasets such as Multinews (56K), it lags behind other models on small out-of-domain datasets like WCEP (10K) and Rotten Tomatoes (3K).

---

### Meta-Review · Area_Chair_gtw3 · 2023-10-02

**Recommendation:** 4

**Metareview:**

The paper proposes a new model which uses Hierarchical context for the task of MDS. The hierarchical schema is designed not only for encoder, where one reviewer pointed with limited novelty, but also for decoder which is interesting. Based on the performance, almost all the reviewers agree that the method competes well with the best-performing models in the field, including those with additional MDS pre-training or with more model parameters. Therefore, I would suggest the paper as Findings in the conference, or  Main conference is also acceptable.

---

### Decision · Program_Chairs · 2023-10-07

**Decision:**

Accept-Findings

**Comment:**

The paper proposes a new model which uses Hierarchical context for the task of MDS. The hierarchical schema is designed not only for encoder, where one reviewer pointed with limited novelty, but also for decoder which is interesting. Based on the performance, almost all the reviewers agree that the method competes well with the best-performing models in the field, including those with additional MDS pre-training or with more model parameters. Therefore, I would suggest the paper as Findings in the conference, or  Main conference is also acceptable.